# Cross-National Variations in COVID-19 Mortality: The Role of Diet, Obesity and Depression

**DOI:** 10.3390/diseases9020036

**Published:** 2021-05-06

**Authors:** Ravi Philip Rajkumar

**Affiliations:** Department of Psychiatry, Jawaharlal Institute of Postgraduate Medical Education and Research (JIPMER), Puducherry 605006, India; ravi.psych@gmail.com; Tel.: +91-413-2296280

**Keywords:** COVID-19, mortality rate, depression, obesity, sugar consumption, seafood consumption

## Abstract

Background: The COVID-19 pandemic has been characterized by wide variations in mortality across nations. Some of this variability may be explained by medical comorbidities such as obesity and depression, both of which are strongly correlated with dietary practices such as levels of sugar and seafood consumption. Methods: COVID-19 mortality indices for 156 countries were obtained from the Johns Hopkins University’s data aggregator. Correlations between these variables and (a) per capita consumption of sugar and seafood, and (b) country-wise prevalence of depression and obesity were examined. Results: Sugar consumption (*r* = 0.51, *p* < 0.001) and prevalence of obesity (*r* = 0.66, *p* < 0.001) and depression (*r* = 0.56, *p* < 0.001) were positively correlated with crude mortality rates, while seafood consumption was negatively correlated with the infection fatality rate (*r* = −0.28, *p* = 0.015). These effects were significant even after correcting for potential confounders. The associations with depression and obesity remained significant upon multivariate regression. Conclusions: Both obesity and depression, which are associated with inflammatory dysregulation, may be related to cross-national variations in COVID-19 mortality, while seafood consumption may be protective. These findings have implications in terms of protecting vulnerable individuals during the current pandemic.

## 1. Introduction

The global pandemic of COVID-19, an acute respiratory illness caused by the novel betacoronavirus SARS-CoV-2, is the defining global health crisis of our times. As of 20 April 2021, over 140 million cases of COVID-19 have been reported globally, and the disease has been directly responsible for over 3 million deaths [1]. Though the majority of COVID-19 infections result in mild or moderate illness, this disease has been associated with a significantly elevated risk of mortality, particularly in the elderly and in those with medical comorbidities [2]. Broadly speaking, COVID-19 is associated with fatalities at a rate that is higher than that of seasonal influenza, but lower than that of the earlier coronavirus outbreaks of Severe Acute Respiratory Syndrome (SARS) and Middle East Respiratory Syndrome (MERS) [3]. Currently, the global COVID-19 crude mortality rate is approximately 39 per 100,000 population, and the median case-fatality ratio—defined as the ratio of deaths to total infections—is estimated to be around 1.8% [1,3,4]. However, from the very first stages of the pandemic, significant variations in mortality have been observed across countries. Initial reports suggested that highly developed European countries with aging populations had higher mortality rates, while Asian and Eastern European countries had fewer deaths [5]; however, as the pandemic has spread and evolved at a global level, a more complex picture has emerged, with lower fatality rates observed in certain Asian and African countries, and higher rates recorded in particular European and South American countries [6,7,8].

Several explanations have been advanced to account for this phenomenon. At the level of the infectious agent itself, it has been suggested that variant strains are associated with a slightly higher risk of death [9]. A wide range of host factors have been associated with a significant increase in COVID-19 mortality, including genetic variants that affect the host immune response [10,11], age [7,12], gender [12] and the presence of comorbid cardiovascular, endocrine, renal or pulmonary disease [13,14]. At a population level, environmental factors such as air pollution [15] and social factors such as the promptness in implementing social distancing measures and the level of adherence to them [7,16], have been associated with variations in COVID-19 mortality.

Recently, evidence has emerged suggesting a significant link between the presence of comorbid depression and an increased risk of mortality. This association has been observed in hospitalized patients [17], in retrospective analyses of health records [18,19] and in ecological analyses of population-level data [13]. The consistency of this association across different study designs and populations suggests that it is unlikely to be a chance finding. Both behavioral and biological mechanisms may mediate this association. For example, depression is often associated with non-adherence to treatment for medical conditions [20,21], as well as “wishes to die” which may lead to a neglect of COVID-19 safety precautions [22]. On the other hand, there is increasing recognition that depression is, in a certain sense, an inflammatory disorder, characterized by significant alterations in immune function, inflammatory activity and oxidative stress [23,24,25], and is genetically linked to alterations in several key immune response genes [26]. Certain inflammatory markers known to be elevated in persons with depression, such as interleukin-6 (IL-6) and interleukin-10 (IL-10) have been associated with disease severity and mortality in patients infected with SARS-CoV-2 [27].

In a similar manner, evidence has emerged linking obesity with increased mortality in patients with COVID-19 [28,29]. This association appears to be consistent across countries, and is significant even after controlling for associated conditions such as diabetes mellitus or systemic hypertension [12]. Moreover, there appears to be a dose–response relationship, with more severe degrees of obesity being associated with worse outcomes [30]. Like depression, obesity is associated with chronic low-grade inflammation, and it has been hypothesized that this mechanism is responsible for the adverse outcomes seen in obese COVID-19 patients [31,32]. These findings are of note because there is a significant mechanistic and clinical overlap between obesity and depression. These disorders may share genetic vulnerabilities [33,34]; several endocrine and inflammatory biomarkers are common to both conditions [35], and depression and obesity are highly comorbid [36]. Recent research has highlighted the possible shared origins of obesity and mood disorders, arising from an evolutionary mismatch between past and modern living environments leading to endocrine and immune–inflammatory dysregulation [37,38]. Thus, the presence of these conditions in combination may have an additive effect on disease severity and mortality in patients with COVID-19.

Though both depression and obesity are partly genetically determined, they share a number of environmental risk factors, particularly diet. While associations between specific dietary patterns and these disorders have been inconsistent [39], stronger evidence exists for links with specific dietary components. In particular, the level of dietary consumption of refined sugar has been associated with the emergence of obesity from childhood onwards [40,41], and there is epidemiological and experimental evidence linking sugar consumption and depression [42,43]. On the other hand, the consumption of fish or seafood, which is rich in omega-3 polyunsaturated fatty acids, appears to be protective against depression [39,42,44], and has also been associated with decreases in peripheral inflammatory markers and successful weight loss in obese individuals undergoing dietary restriction [45].

There is little research on the impact of dietary factors on COVID-19 [46], and to date, the interaction between specific dietary components and comorbid medical conditions in terms of effects on COVID-19 mortality has not been specifically examined. Recommendations regarding diet-based interventions for COVID-19 need to be supported by evidence [47]. Thus, the current study was designed to assess the impact of two specific dietary components (sugar and fish) and two closely linked comorbid diagnoses (depression and obesity) on two widely used indices of COVID-19 fatality: the crude mortality rate and the case-fatality ratio, using country-level data from reliable sources.

## 2. Materials and Methods

### 2.1. Data Sources

Information on indicators of COVID-19 fatality were obtained from the Johns Hopkins University’s Coronavirus Resource Center, which provides continuously updated information on the numbers of new cases and deaths for each country and region based on data from regional and local health authorities [1]. Two indicators of deaths due to COVID-19 are available from this resource:

The crude mortality rate, defined as the number of deaths per 100,000 population. This indicator provides a broad index of the impact of COVID-19 on the general population as a whole in terms of mortality. It has been widely used in prior ecological research on factors associated with COVID-19 mortality [4,13]; however, it is significantly affected by the total population size (the denominator), as well as local practices in attributing deaths to COVID-19 [48];The case-fatality ratio, defined as the ratio of deaths to infections and expressed as a percentage. This indicator provides an estimate of what proportion of patients with COVID-19 will have a fatal outcome. This index has also been used extensively in ecological studies of the COVID-19 pandemic [6,7] and has the advantage of not being directly affected by population size. However, it is significantly affected by the number of tests carried out in the general populations; low-income countries may have artificially high case-fatality ratios because they lack the resources to test asymptomatic or mild cases [13,48].

Data on these two indices were available for a total of 156 countries. For the purpose of this study, the estimated values of these two indices as of 31 March 2021 were used.

Information on the prevalence of depression for each country was obtained from the World Health Organization (WHO)’s publication Depression And Other Common Mental Disorders: Global Health Estimates, which provides estimates of the point prevalence of depression for all nations based on the Global Health Estimates for the year 2015 [49]. In order to assess the accuracy of these data, the prevalence obtained from this dataset were compared with data obtained from the World Mental Health Survey, which was a rigorous epidemiological study of the point and lifetime prevalence of depression in 18 developed and developing countries [50]. There was a strong positive correlation between the Global Health Estimates and the World Mental Health Survey data (Pearson’s *r* = 0.7, *p* = 0.012, df = 16), suggesting that these estimates are reliable.

Information on the prevalence of obesity was obtained from the WHO Global Health Observatory data on risk factors for non-communicable diseases, which provide estimates of the point-prevalence of obesity in each country [51].

Information on the per capita consumption of sugar and seafood was retrieved via separate database queries from the Food and Agriculture Organization’s Corporate Statistical Database (FAOSTAT), which collects and distributes statistical data on food production and consumption from the year 1961 onwards [52]. For this study, the most recent estimates of consumption were recorded, which were available for the year 2018 for sugar and 2013 for seafood. Data on sugar consumption was measured in kilograms per capita per year, and was available for 156 countries, while data on seafood consumption was measured in grams per capita per day and was available for 155 countries.

### 2.2. Measurement of Potential Confounders

As the current study is based on ecological data, corrections for potential confounding factors are essential. In view of the variables being included for analysis in this study, the following parameters were included as potential confounders for each country:
National life expectancy, in view of the robust association between advanced age and COVID-19 mortality. Data on this variable was obtained from the official statistics of the World Bank [53];Estimated prevalence of diabetes mellitus, as this condition is independently associated with COVID-19 mortality and is often comorbid with both depression and obesity [54]. Data on this variable was obtained from the aggregates of the International Diabetes Federation and Diabetes Atlas data, available at the World Bank website [55];Number of hospital beds per 100,000 population, as hospital bed availability has been identified as an independent predictor of case fatality [12]. Data on this variable was obtained from the United Nations’ Human Development Report for the year 2020 [56].

### 2.3. Data Analyses

All variables were tested for normality using the Kolomogorov–Smirnov test. Both outcome measures (COVID-19 crude mortality rate and case fatality ratio) did not show a normal distribution, with plots of the distribution showing positive skewness in both cases. In view of this, both these variables were transformed to conform to a Gaussian distribution by taking the natural logarithm of the raw data for each variable. Pearson’s correlation coefficient (*r*) was used to test for a significant linear relationship between these indices and the prevalence of depression and obesity, as well as with the per capita consumption of fish and sugar. All significance values were corrected for multiple comparisons using Bonferroni’s method, to minimize the possibility of false-positive associations arising from multiple tests. A corrected significance level of *p* < 0.05 was considered significant for bivariate analyses.

In order to ensure that these associations were not due to the confounding effect of age, diabetes mellitus or hospital bed availability, the relationship of these three variables with the outcome measures was also assessed using Pearson’s correlation for log-transformed values, with corrections for multiple comparisons as above. Potential confounders identified as significant at *p* < 0.05 or less were then included in a partial correlation analysis, to examine whether the results of direct bivariate correlations remained significant when correcting for these factors.

Finally, in order to assess the relative contributions and significance of these associations, multivariate linear regression analyses were carried out using the natural logarithms of the crude mortality rate and case-fatality ratio as dependent variables, and all variables identified as significant at *p* < 0.05 (corrected) in the bivariate analyses as independent variables. These analyses were only carried out if more than one variable was significantly associated with the specific outcome.

## 3. Results

### 3.1. Bivariate Analyses

Scatter plots of the distributions of dietary component consumption and specific comorbidities against the crude mortality rate and infection-fatality ratio are presented in Figure 1 and Figure 2, respectively. It may be inferred from these figures that possible linear or monotonic associations with the crude mortality rate exist for sugar consumption and the prevalence of depression and obesity, but not for seafood consumption. On the other hand, no such association could be inferred between any of these variables and the case-fatality ratio. The crude mortality rate and case-fatality ratio were positively correlated with each other, but the strength of this association was relatively weak (*r* = 0.275, *p* < 0.01 for log-transformed values). No significant correlation between national population and crude mortality rates for COVID-19 was identified (*r* = −0.123, *p* = 0.126 for log-transformed crude mortality rate), suggesting that the confounding effect of population size on this index is likely to be small.

Results of the bivariate analyses for the chief study variables are presented in Table 1. For the COVID-19 crude mortality rate, moderately strong positive correlations were observed with per capita sugar consumption (*p* < 0.001), prevalence of obesity (*p* < 0.001) and prevalence of depression (*p* < 0.001), with the strongest correlation being observed for the prevalence of depression. No significant relationship was observed between per capita seafood consumption and crude mortality rates per country.

For the COVID-19 case-fatality ratio, a weak negative correlation was observed with per capita seafood consumption, which remained significant after Bonferroni’s correction (*p* = 0.015). No significant correlations were observed between this index and sugar consumption or the prevalence of depression or obesity.

Though positive correlations between the prevalence of depression and obesity (Pearson’s *r* = 0.64, *p* < 0.001), and between these variables and sugar consumption (Pearson’s *r* = 0.68 for sugar x obesity and 0.52 for sugar x depression, *p* < 0.001 for both correlations) were observed, these were moderate in strength, suggesting that there was unlikely to be significant multicollinearity between these variables.

### 3.2. Analyses of Potential Confounders

On direct bivariate analyses of potential confounders, it was found that life expectancy (*r* = 0.55, *p* < 0.001) showed a moderate positive correlation with the log-transformed COVID-19 crude mortality rate, while hospital bed strength (*r* = 0.163, *p* = 0.30) and the prevalence of diabetes mellitus (*r* = 0.119, *p* = 0.84) showed non-significant positive correlations with this index. None of these variables were significantly correlated with the transformed case-fatality ratio.

Based on these results, a set of partial correlation analyses were carried out to examine whether the relationships observed for the COVID-19 crude mortality rate remained consistent when controlling for the confounding effect of life expectancy. In these analyses, a weak positive relationship between sugar consumption and the log-transformed crude mortality rate remained, even after correcting for multiple comparisons for a 4 × 4 table (partial *r* = 0.26, *p* = 0.02), while somewhat stronger positive relationships were confirmed for both the prevalence of depression (partial *r* = 0.37, *p* < 0.001) and the prevalence of obesity (partial *r* = 0.47, *p* < 0.001). These results suggest that the impact of these variables—and particularly of the comorbid diagnoses—remained substantial even when other risk factors for COVID-19 mortality were taken into account.

Though a significant relationship between seafood consumption and the crude mortality rate was not observed on univariate analysis, a secondary analysis was carried out examining the relationship between these two variables when correcting for sugar consumption as well as for life expectancy. This is because prior research has shown that the presence of other dietary components may nullify the positive health effects of seafood consumption [57]. In this analysis, a weak negative correlation was observed between per capita seafood consumption and the COVID-19 crude mortality rate (partial *r* = −0.19, *p* = 0.021).

### 3.3. Multivariate Analysis

To better delineate the relative contributions of the various factors associated with COVID-19 mortality, a multivariate linear regression analysis was carried out using the logarithm of the COVID-19 crude mortality rate as the dependent variable, and entering all the variables found to be significantly associated with this index on bivariate analyses—sugar consumption, prevalences of depression and obesity and life expectancy—as independent variables. As only a single variable—per capita fish consumption—was associated with the case-fatality ratio even in uncorrected analysis, multivariate analysis was not attempted for this index.

The results of this analysis are provided in Table 2.

The model accounted for around 47% of the variance in the COVID-19 crude mortality rate (R^2^ = 0.484; adjusted R^2^ = 0.471). Three individual variables—point prevalence of obesity (β = 0.41, *p* < 0.001), point prevalence of depression (β = 0.19, *p* = 0.017) and life expectancy (β = 0.17, *p* = 0.041)—retained their significance in this model, and sugar consumption showed a non-significant positive correlation. The variance inflation factors for each variable ranged from 1.83 to 2.54. As all these values were less than 4, there was no evidence of significant multicollinearity between these variables.

### 3.4. Exploration of the Effect of Outliers, and of Possible Non-Linear Relationships

Visual inspection of the scatter plots revealed a potential confounding effect of outliers for the level of seafood consumption, with very high values (>100 kg/capita/year) being reported in two island countries, the Maldives and Papua New Guinea. Omission of these two values led to a reduced significance for the association between seafood consumption and the log-transformed case-fatality ratio (*r* = −0.197, *p* = 0.014).

As earlier research has suggested that non-linear relationships might exist between dietary components and health outcomes in other disorders [58,59], an attempt was made to test for the possibility of such relationships using the SPSS curve estimation function. It was found that a cubic model might provide a slightly better fit of the relationship between seafood consumption and case-fatality ratios, particularly after removing outliers (R^2^ = 0.05 for the cubic model, against 0.04 for the linear model; *p* = 0.048). A similar effect was observed for the relationship between sugar consumption and crude mortality rates, where cubic (R^2^ = 0.38) and quadratic (R^2^ = 0.37) curves fit the observed data better than a linear model (R^2^ = 0.25). This supports the prior observations that non-linear models might better explain the relationship between diet and adverse health outcomes.

## 4. Discussion

The results of this study suggest that significant positive relationships exist between the point prevalence of depression and obesity and the COVID-19 crude mortality rate across countries. These results are consistent with the results of earlier research which has found that both these disorders confer an added risk of mortality in COVID-19 patients [13,17,18,19,28,29]. On bivariate analyses, sugar consumption was also associated with the crude mortality rate; however, on multivariate analyses, this relationship, unlike the former two, was no longer statistically significant. This finding suggests that any harmful effect of sugar consumption is indirect, and probably mediated through its association with the subsequent risk of obesity [40,41] and depression [42,43] at a population level. The effects of these factors may be additive with other risk factors for mortality. For example, the results of the multivariate analysis suggest that advanced age and depression are both associated with mortality in patients with COVID-19; this was corroborated by a recent study of 122 elderly adults showing that depression was associated with increased mortality in this age group [60].

### 4.1. Mechanisms Linking Depression and Obesity with COVID-19 Mortality

Clues regarding the mechanisms linking depression and poor outcomes in COVID-19 may be inferred from observational studies. For example, a recent study in a large Spanish sample of COVID-19 inpatients (*n* = 2150) found that a history of depression, but not the presence of de novo depressive symptoms, was associated with mortality [61]. This suggests that the pathways linking depression and increased mortality are not necessarily related to the current presence or severity of depressive symptoms, but to factors of a longer duration. Foremost among these potential mechanisms are alterations in immune response, which may persist even after a patient’s depressive symptoms have improved [62]. In older adults, elevations in IL-6 and C-reactive protein (CRP) have been documented even prior to the onset of syndromal depression [63,64], suggesting that they are a “trait” rather than a “state” marker of depression. Similarly, the neutrophil-to-lymphocyte ratio, a marker of systemic inflammation, is elevated in patients with major depression even after antidepressant treatment [65,66], and this parameter has been identified as a predictor of COVID-19 mortality in prospective studies of hospitalized patients [67,68]. Second, recurrent or chronic depression is associated with a sedentary lifestyle characterized by less physical activity or exercise [69], less healthy dietary practices [70] and the use of substances, particularly alcohol [71]. These lifestyle factors may independently increase COVID-19 mortality [72]. Third, depression is frequently comorbid with conditions such as diabetes mellitus, cardiovascular, pulmonary and renal disease [73], which are associated with worse outcomes in COVID-19 [13,14]. Finally, depression is associated with delays in seeking diagnostic testing or treatment for medical conditions [74], which may be both due to the reduced activity levels and motivation that are part of depression [75] and the stigma associated with this condition, serving as a barrier to health care access [76].

In the multivariate analysis, the effect size and significance level for obesity in influencing the COVID-19 crude mortality rate were greater than for depression or age, suggesting that the contribution of this factor to variations in mortality may be substantial. There are several factors that may explain the association between obesity and increased mortality in COVID-19, either in whole or in part [77]. The first of these is the broad range of metabolic alterations seen in many individuals with obesity, including alterations in inflammatory activity, lipid, hepatic enzyme and endocrine profiles [78]. As these changes arise from a distinct pathway involving the release of specific molecules from adipose tissue, they may act synergistically with the changes seen in depression to enhance adverse outcomes [79]. Second, angiotensin-converting enzyme (ACE-2), which is highly expressed in adipose tissue, acts as a receptor for the entry of SARS-CoV-2; this may lead to an increased risk of infection as well as of severe outcomes in patients with COVID-19 [80]. Third, obesity is associated with vascular dysfunction, particularly altered endothelial function and accelerated atherosclerosis [81]. These changes predispose patients to the development of cardiovascular disease, which in turn has an adverse impact on COVID-19 mortality [13,82]. Obesity may also cause respiratory difficulties, and may lead to practical problems when maintaining patients on assisted ventilation with higher degrees of obesity; however, evidence regarding the importance of this factor has been inconsistent [83,84]. Finally, like depression, obesity is characterized both by unhealthy lifestyle patterns, particularly with regard to physical activity [76], and with a high level of shame and stigma leading to delays in accessing essential healthcare [85].

From the above, it is clear that the mechanisms linking obesity and depression to poor outcomes in COVID-19 overlap to a large extent. A more careful analysis of some of these factors would require the analysis of other parameters, such as rates of other medical comorbidities, or laboratory markers of immune or cardiovascular dysfunction. As it was not possible to assess for these factors in an analysis of this kind, a true estimate of their importance would require careful assessment in retrospective and prospective clinical samples.

### 4.2. Diet as an Indirect Mediator of Mortality in COVID-19

The results of this study suggest that the effect of diet on outcome in COVID-19 is likely to be indirect, and mediated through the presence of comorbid conditions associated with particular dietary patterns and practices. Alternately, diet may act through other mechanisms, such as alterations in systemic inflammatory activity, in a way that is partly independent from the presence of these comorbidities. Earlier research has suggested that sugar consumption is associated with elevated crude mortality rates in COVID-19, which is consistent with this study’s findings, while the consumption of beans and legumes may have a protective effect; however, these analyses were not corrected for any potential confounders [46]. Though evidence for a protective effect of seafood consumption on measures of COVID-19 fatality was found in this study, the magnitude of this effect was small. This suggests that the effect of this dietary component on outcomes is confounded by other factors, which include other dietary components, age and the presence of medical comorbidities. However, a systematic review of interventional studies suggests that strategies to improve dietary quality in patients with obesity, with or without depression, may improve both physical and mental health parameters [86]. Furthermore, interventions involving dietary components may partially improve the functioning of physiological pathways that are dysregulated in both these disorders [87]. Thus, even if an ecological analysis of this kind demonstrates only modest effects, dietary or nutrient-based strategies may have significant effects at the population level, particularly in high-risk individuals who have either of the comorbidities examined in this paper [88], as they may act through a wide range of molecular pathways to minimize inflammatory and endothelial dysfunction [89]. Further, as suggested by the results in Section 3.4, the relationship between diet and COVID-19 outcomes may be non-linear. Recommendations to minimize sugar consumption and enhance seafood consumption may be beneficial, though their overall effect may be modest.

### 4.3. Implications for Preventive and Mitigation Strategies

A complete review of integrated prevention strategies for individuals with obesity or depression in the context of the COVID-19 pandemic is beyond the scope of this paper. However, potential strategies of interest may include: the provision of information regarding COVID-19 preventive measures in a clear manner, without causing undue alarm [90,91]; the provision of brief psychological interventions and lifestyle advice using remote technologies or mobile applications [92]; the initiation or continuation of antidepressant therapy for patients with depression, which may have antiviral properties over and above its beneficial psychological effects [93,94,95]; and the use of dietary or nutraceutical-based interventions for those with obesity [96]. Considering the higher risk of adverse outcomes in patients with these disorders, they should be encouraged to receive immunization and assisted in accessing it; however, a note of caution is justified given the concerns for reduced host immune responses in these disorders [97,98]. More detailed discussions of some of these strategies are available in published reviews [99,100].

### 4.4. Methodological Issues and Limitations

The interpretation of the results of this study is subject to certain methodological considerations. First, the indicators used to estimate mortality were obtained from aggregates of public reports, and may be sensitive to factors such as under-reporting (particularly in low- and middle-income countries) and local practices involved in attributing deaths to COVID-19 [3,5,6]. Second, estimation of case fatality rates is critically dependent on the number of mild or asymptomatic cases diagnosed, significantly impairing the sensitivity of this indicator [44,46]. Some researchers have suggested the use of the excess mortality due to COVID-19 as a better measure of mortality; however, information on this variable is not widely available. Third, estimates of dietary components are based on estimates and calculations for each country, and may fail to reflect regional variations within a country, such as increased seafood consumption in coastal areas. Fourth, though an attempt was made to correct for potential confounding factors, several other potentially relevant variables, such as other dietary components, socioeconomic and cultural factors and medical conditions, were not studied. Fifth, the data used in this study may be outdated to some extent, particularly the information on seafood consumption which was last recorded for the year 2013. Finally, ecological analyses can only identify correlations and possible associations; confirmation of the possibilities raised by this study would require clinical research in hospital and community samples.

## 5. Conclusions

Though the results presented here are subject to certain methodological limitations, they are biologically plausible and consistent with the existing literature, both in hospitalized samples and at a cross-national level, as well as with the existing hypothesis that obesity and associated conditions can account for population and ethnic differences in COVID-19 outcomes [80]. The implementation of strategies to aid patients with these chronic conditions during the COVID-19 pandemic could lead to a significant reduction in mortality. In the long term, the adoption of healthy lifestyle practices, particularly related to diet and physical activity, as well as the effective treatment of depression and obesity at an early stage, could attenuate the impact of future waves of this pandemic, as well as build resilience to future disease outbreaks at a population level. Further investigation of the roles of antidepressant therapy, changes in dietary composition, specific nutritional components and behavioral strategies aimed at promoting lifestyle change during the pandemic are warranted in the light of these findings.

## Figures and Tables

**Figure 1 diseases-09-00036-f001:**
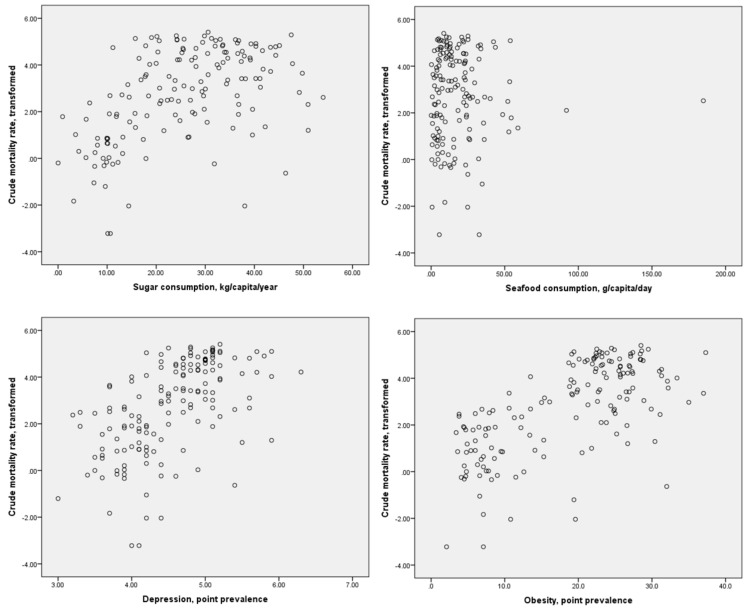
Scatter plots of sugar and seafood consumption and prevalence of depression and obesity against log-transformed crude mortality rates for COVID-19.

**Figure 2 diseases-09-00036-f002:**
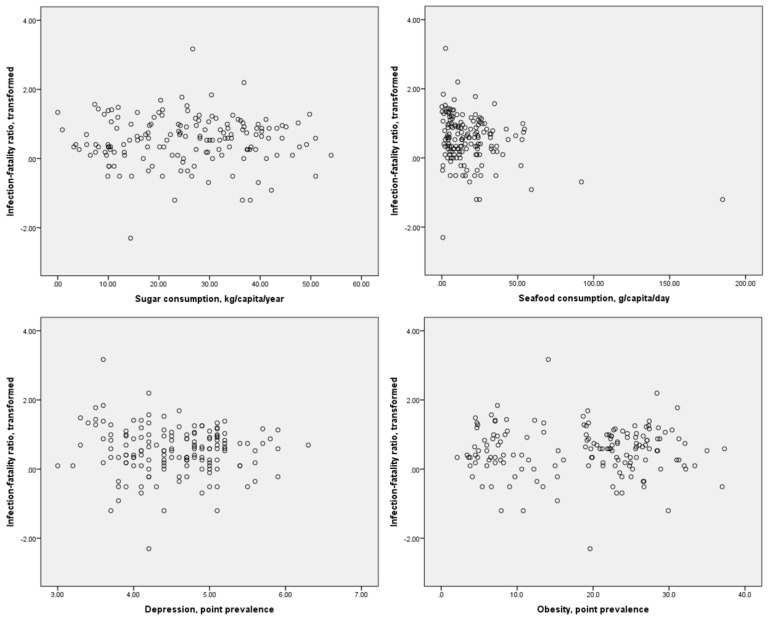
Scatter plots of sugar and seafood consumption and prevalence of depression and obesity against log-transformed case-fatality ratios for COVID-19.

**Table 1 diseases-09-00036-t001:** Bivariate correlations between diet, obesity, depression and COVID-19 fatality measures.

Variable	COVID-19 Crude Mortality Rate	COVID-19 Case Fatality Ratio	Sugar Consumption, kg/Capita/Year	Seafood Consumption, g/Capita/Day	Depression, Point Prevalence (%)	Obesity, Point Prevalence (%)
COVID-19 crude mortality rate	*	0.275 (0.018) **	0.51 (<0.001) **	<0.01 (0.999)	0.56 (<0.001) **	0.66 (<0.001)
COVID-19 case-fatality ratio	-	*	0.01 (0.999)	−0.28 (0.015) **	0.01 (0.999)	−0.07 (0.999)
Sugar consumption	-	-	*	0.07 (0.999)	0.52 (<0.001) **	0.68 (<0.001) **
Seafood consumption	-	-	-	*	0.34 (0.999)	0.07 (0.999)
Depression, point prevalence	-	-	-	-	*	0.64 (<0.001) **

Note: All correlations are given in the form: Spearman’s correlation coefficient (significance level). All significance values are corrected for a 6 × 6 correlation matrix. All correlations are for log-transformed crude mortality rates and case-fatality ratios. *, value omitted (same variable in row and column); **, significant at *p* < 0.05 after Bonferroni’s correction.

**Table 2 diseases-09-00036-t002:** Multivariate linear regression analysis of the predictors of COVID-19 crude mortality rate (log-transformed).

Variable	Correlation Coefficient (β)	Significance Level	Part Correlation	Variance Inflation Factor
Sugar consumption	0.03	0.739	0.02	2.04
Depression, point prevalence	0.19	0.017	0.14	1.83
Obesity, point prevalence	0.41	<0.001	0.26	2.54
Life expectancy	0.17	0.041	0.12	1.96

## Data Availability

The data that were used for this study are available from public domain resources that are listed in references [49,51,52,53,55,56]. A complete data sheet including all study variables is available from the author upon reasonable request.

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
