# Peer review of "Cross-National Variations in COVID-19 Mortality: The Role of Diet, Obesity and Depression"

_diseases, 2021, doi:10.3390/diseases9020036_

Round 1

Reviewer 1 Report

Manuscript ID: diseases-1216925

Ravi Philip Rajkumar: Cross-national variations in COVID-19 mortality: the role of 2 diet, obesity and depression

I enjoyed reading this well-written manuscript of associations between obesity, depression, and the incidence of COVID-19. I have just a couple of minor comments on the paper.

Lines 75-77: We have explained links between obesity and the rate of COVID-19 disease in detail in some other paper published by MDPI:

Krams et al. 2020. COVID-19: fat, obesity, inflammation, ethnicity, and sex differences. – Pathogens 9(11): 887.

Lines 83-86: I would suggest the author read our three recent papers explaining associations between specific dietary patterns and disorders linked with low-grade inflammation of the central nervous system:

Luoto et al. 2018. Depression subtyping based on evolutionary psychiatry: From reactive short-term mood change to depression. – Brain, Behavior and Immunity 69: 630.

Rantala et al. 2021. Bipolar disorder: An evolutionary psychoneuroimmunological approach. – Neuroscience & Biobehavioral Reviews 122: 28–37; Hope these can help!

Line 161: “2.3. Data analysis” should be changed to “Data analyses” because the author performed several types of analyses.

Line 170 and throughout the text: I am not sure about the Diseases requirements but usually “p” in p-values are italicized.

Lines 219-220: Since p = 0.11, I would recommend deleting “though a marginal negative correlation was observed for hospital bed 219 strength (ρ = -0.13, p = 0.11)”.

I. Krams

Author Response

Thank you very much for your thoughtful review of my manuscript.

A detailed response to your comments, outlining the corrections made, is attached here.

Reviewer 2 Report

This paper reports an interesting new analysis of cross-national data and its focus on diet adds important new information. It is very up to date given the March 31st 2021 data collection and this review being conducted 28th April.  I have the following observations in no particular order of importance:

  • The use of non-parametric Spearman’s correlations is suitably cautious but the author then goes on to conduct a log transformation of the crude mortality rate to allow a parametric regression analysis. Given that the transformation appears to render the mortality variable in a normal form (I am assuming it does), why not present parametric correlations throughout using the transformed variables?
  • I felt that the data would lend themselves to graphical presentation. Scatterplots, and using these to identify outlier countries could be quite informative.  There is nothing here about ‘odd’ countries that do not fit the simple monotonic model.   Notwithstanding the above log transformation, it might have been informative to explore some non-linear curve fitting.  It would be good to ‘see’ the data.
  • In reporting the multiple regression, the part correlations would be informative.
  • The correlation between crude mortality rate and seafood consumption in Table 1 is given as 0.07, p = 0.417 yet the correlation has two asterisks indicating p<0.05 after correction. The bracketed p-values are supposed to be the corrected ones?
  • The correlation between seafood consumption and fatality rate is given as -0.19, p = .016 in the abstract but this correlation is not significant after the correction as reported in Table 1. IS this effect big enough to warrant inclusion in the abstract?
  • Might it be better to present the correlations between all variables in Table 1 – i.e. the full 6x6 matrix?
  • When doing the Bonferroni corrections would it not be easier to follow if a revised alpha-criterion was specified rather than presenting two p-values for each relationship?
  • The Discussion section, whilst very readable and informative, moved quite a long way from the data that is being presented. For e.g., the implications for mitigation section could probably be omitted given that these implications follow from other studies that have already reported on the importance of public health interventions to address diet and lifestyle.
  • The limitations section is fine but could do with a mention of the time lag between the measures. The seafood data, for e.g., dates from 2013.

Author Response

Thank you very much for your in-depth review of my paper.

I have incorporated all the corrections you have suggested, and these are described in detail in the attachment.
